# Towards Multi-Domain Chinese Document VQA: a New Dataset and Baseline Method

## Abstract

Document Visual Question Answering (DocVQA) remains a significant challenge in the field of document understanding and is a critical evaluation metric for current general-purpose large model techniques. However, the prevailing public datasets are predominantly designed for single scenarios or specific sources. Furthermore, most available datasets are in English, which limits the verification of model performance in other languages. This paper presents a novel multi-domain Chinese document VQA dataset, which includes 39 document types from 7 different domains. The designed question set encompasses both common extractive questions and complex abstractive questions. Based on this dataset, we conducted a comprehensive review and analysis of various technical paradigms, including both traditional and large model-based approaches. Using the popular in-context learning framework, we propose a strong baseline that achieves commendable few-shot adaptation. Comparative evaluations demonstrate the superior performance of the proposed method across different solution paradigms. The dataset and code will be published.

## 1 Introduction

Document understanding plays a central role in artificial intelligence, covering various industries and enhancing daily operational efficiency. Previous endeavors have primarily focused on distinct subfields driven by different task objectives, encompassing document classification (Mohbat et al., 2023; Fronteau et al., 2023), key information extraction (KIE) (Zhang et al., 2020; Tang et al., 2021; Wang et al., 2021a), layout analysis (Zhang et al., 2021; Cheng et al., 2023; Shen et al., 2021), table understanding (Shigarov, 2023; Li et al., 2022), *etc*. Recently, the advent and proliferation of large models have instigated a shift towards a unified paradigm for end-to-end problem representation. Document Visual Question Answering (DocVQA) (Mathew et al., 2021) theoretically aligns with such a paradigm, potentially addressing more intricate task demands compared to the aforementioned objectives.

Traditional approaches of DocVQA generally fall into two categories: those relying on pure language models (Tito et al., 2022) and those leveraging multimodal pretrained models (Xu et al., 2021a; Huang et al., 2022; Peng et al., 2022; Appalaraju et al., 2021). The latter, integrating additional modalities such as vision and layout, demonstrate superior performance. The recent emergence of Large Language Models (LLMs) and Large Vision-Language Models (LVLMs) has introduced innovative solutions. For example, research such as Liu et al. (2023b); Yang et al. (2023b) incorporates off-the-shelf Optical Character Recognition (OCR) results into the prompt and utilizes LLMs as information extractors. LVLM-based approaches directly integrate OCR capabilities into the model, offering a pure end-to-end solution (Alayrac et al., 2022; Yang et al., 2023a; Anil et al., 2023; Anthropic, 2024; Chen et al., 2023; Zhu et al., 2023; Bai et al., 2023b; Li et al., 2023; Wang et al., 2023; Zhang et al., 2023), which exhibits promising potential in text-oriented task evaluation. However, most existing models do not support Chinese document comprehension due to the lack of corresponding training and evaluation data.

Presently available datasets of DocVQA predominantly originate from homogeneous domains and are primarily in English, restricting method generalizability to diverse scenarios. For example, the images from Mathew et al. (2021) are mostly scanned industrial documents, VQA-CD (Mahamoud et al., 2022) focuses only on invoice scenarios, and both TAT-DQA (Zhu et al., 2022) and BD-VQA

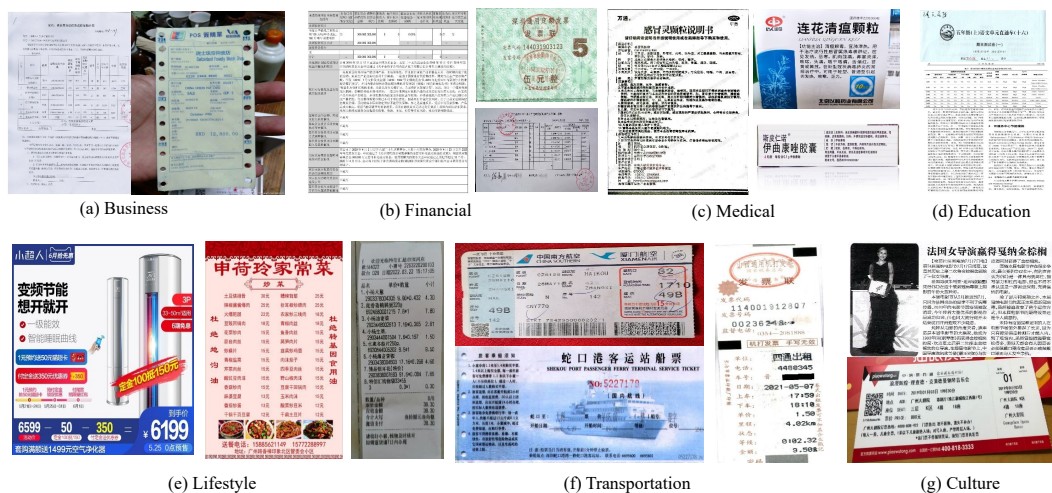

(a) Business      (b) Financial      (c) Medical      (d) Education

(e) Lifestyle      (f) Transportation      (g) Culture

Figure 1: Samples from multiple domains in MDCD-VQA dataset.

(Raja et al., 2023) come from financial reports. Chinese, as a crucial language spoken by the largest population worldwide, presents additional challenges due to its extensive character set (Shi et al., 2023a). While Qi et al. (2022) pioneers a Chinese DocVQA dataset, all images are sourced from screenshots of the webpage, and questions are only posed in an extractive way.

This paper introduces a novel Multi-Domain Chinese Document Visual Question Answering (MDCD-VQA) dataset, which aims to encompass a broad spectrum of document types across various real-world domains. The construction of the dataset involved the aggregation of data from multiple public and private databases. In contrast to previous datasets, which were predominantly composed of documents with regular text (*e.g.*, extracted from PDF or web pages), MDCD-VQA incorporates a significant amount of real-world data sourced from photographed or scanned scenes. Consequently, these introduce perceptual challenges to models like skew, curvature, blur and overlap. The MDCD-VQA dataset comprises 5,071 images and 34,170 questions, and some samples from different domains are shown in Figure 1. The construction of question-answer (Q&A) pairs was guided by rigorous principles to ensure diversity. The dataset encompasses both extractive questions, where the answer is located within the image, and more complex abstractive questions that involving tasks such as summarization, judgment, inference, and calculation.

Using our proposed MDCD-VQA dataset, we conduct a comprehensive evaluation and analysis of various method types for DocVQA tasks. This evaluation encompasses traditional full-training-based methods as well as zero/few-shot-based approaches leveraging LLMs and LVLMs. Furthermore, we introduce a novel baseline method based on the In-Context Learning (ICL) framework (Dong et al., 2023), which integrates powerful LLMs. This model leverages the retrieval of the most similar examples from the image, text, and question perspectives in the training set to activate the underlying capabilities of LLMs. This approach represents a viable option offering optimal comprehensive performance and generalization at the current stage of development.

The main contributions of this paper are as follows:

- We present a new Chinese DocVQA dataset, which, to the best of our knowledge, is the first comprehensive dataset to include both extractive and abstractive questions across multiple domains.

- We conducted extensive experiments to evaluate four different architectural solutions for this Chinese DocVQA task. The results underscore the substantial potential for improvement in Chinese document understanding by current large models.

- We introduce a strong baseline model based on the ICL framework. The experimental results show that this method outperforms previous approaches on the proposed dataset.

| Dataset | Language | Scene Source | Imaging Source | Question Type | #Document | #Q&A pairs |
|---|---|---|---|---|---|---|
| DocVQA (Mathew et al., 2021) | English | Industry documents | Scan | Extractive | 12,767 | 50,000 |
| InfographicVQA (Mathew et al., 2022) | English | Infographics | BD | Extractive/Abstractive | 5,485 | 30,035 |
| VisualMRC (Tanaka et al., 2021) | English | Web pages | BD | Abstractive | 10,234 | 30,562 |
| VQA-CD (Mahamoud et al., 2022) | English | Invoices | Scan | Extractive | 693 | 3,000 |
| DuReaderVis (Qi et al., 2022) | Chinese | Web pages | BD | Extractive | 15,000 | 15,000 |
| DocVQA-ZH [1] | Chinese | Insurance related docs | BD/Scan | Extractive | 5,243 | 40,385 |
| MP-DocVQA † (Tito et al., 2022) | English | Industry documents | Scan | Extractive | 5,928 | 46,176 |
| TAT-DQA† (Zhu et al., 2022) | English | Financial reports | BD | Extractive/Abstractive | 2,758 | 16,558 |
| SlideVQA † (Tanaka et al., 2023) | English | Slide decks | BD | Extractive | 2,600 | 14,500 |
| DUDE † (Van Landeghem et al., 2023) | English | Multi-domain | BD/Scan | Extractive/Abstractive | 5,000 | 41,541 |
| MDCD-VQA (ours) | Chinese | Multi-domain | BD/Scan/Camera | Extractive/Abstractive | 5,071 | 34,170 |

Table 1: Comparisons with existing Document VQA datasets. Dataset with † means the document has multiple pages."#" means "the number of". "BD" is short for "Born Digital".

## 2 RELATED WORKS

### 2.1 DATASETS OF DOCVQA

DocVQA is a subset of text-oriented VQA tasks. Unlike typical tasks where the questions focus primarily on prominent text in the image, as seen in tasks like TextVQA (Singh et al., 2019), OCRVQA (Mishra et al., 2019), and STVQA (Biten et al., 2019), DocVQA images have denser text with more fine-grained associated questions.

Most of the existing public datasets for DocVQA are tailored for specific domains or from specific sources. For example, the datasets presented in Mathew et al. (2021); Tito et al. (2022) are constructed using images selected from the UCSF Industry Documents Library. VQA-CD (Mahamoud et al., 2022) contains invoice images from an industry document collection. TAT-DQA (Zhu et al., 2022) and BD-VQA (Raja et al., 2023) are two datasets within the financial domain, the former consisting of comprehensive financial reports and the latter containing specific financial spreadsheets. VisualMRC (Tanaka et al., 2021) and DuReader$_{vis}$ (Qi et al., 2022) are different English and Chinese datasets for the DocVQA task, mainly containing screenshots of web pages. The collection of SlideVQA (Tanaka et al., 2023) all from slide decks. A more recent addition is the multi-page, multi-domain dataset DUDE (Van Landeghem et al., 2023), proposed and used as a competition dataset for ICDAR-2023 (International Conference on Document Analysis and Recognition). For Chinese datasets, DocVQA-ZH[1] is a competition dataset that contains various types of scanned documents related to insurance scenarios such as medical bills or cases. However, the download for this dataset is currently unavailable due to the conclusion of the competition. Notably, the question types in this dataset are exclusively in extractive form. A comprehensive comparison of these datasets can be found in the table 1.

### 2.2 METHODS OF DOCVQA

Given the denser text information in DocVQA, models need to have enhanced text understanding capabilities. As a result, proposed solutions are predominantly implemented within the language model paradigm. Besides the direct adoption of language models and treating the problem as a QA task (*e.g.,* BERT (Devlin et al., 2019) or BigBird (Zaheer et al., 2020) in Tito et al. (2022)), the dominant approaches in the past few years involve the use of a multimodal pre-trained model. These approaches typically incorporate features from three modalities (visual, layout, semantic) into a transformer-based encoder. Representative works include the LayoutLM family (Xu et al., 2020; 2021a; Huang et al., 2022; Xu et al., 2021b), DocFormer (Appalaraju et al., 2021; 2023), StrucText (Li et al., 2021; Yu et al., 2023b), ERNIE-Layout (Peng et al., 2022),*etc.* Despite the use of off-the-shell OCR results, there have also been some recent works like Donut (Kim et al., 2022), Dessurt (Davis et al., 2022) and Pix2Struct (Lee et al., 2023) aiming to build more end-to-end OCR-free solutions. These systems usually integrate the ability of OCR in the pre-training stage, but the current public models mainly have the ability of English scenes only.

Recently, there has been a surge in the development of general paradigms based on LLMs and LVLMs. These models aim to support different tasks, and some works such as GPT-4 (Yang et al.,

[1]http://ailab.aiwin.org.cn/competitions/49

| Dataset | Source | Scene | Original Task | #Original Images | #Selected Images |
|---|---|---|---|---|---|
| EPHOIE (Wang et al., 2021a) | Public | Examination paper | KIE | 1,494 | 181 |
| EATEN-BC (Guo et al., 2019) | Public | Business card | KIE | 200k | 45 |
| SCID (Qiao et al., 2023) | Public | Financial invoice | KIE | 40,716 | 509 |
| CER-VIR-ZH [2] | Competition | Shopping receipt | KIE | 1,405 | 381 |
| ComFinTab-ZH (Li et al., 2022) | Public | Financial table | Table Understanding | 6,000 | 291 |
| CDLA [3] | Public | Academic literature | Layout Analysis | 6,000 | 318 |
| DI dataset (Li et al., 2020) | Public | E-commerce picture | Reading Order Detection | 7,475 | 515 |
| XFUNSD-ZH (Xu et al., 2021b) | public | Various forms | KIE& Entity Linking | 199 | 196 |
| HUST-CELL (Yu et al., 2023a) | Competition | Multi-scenario | Entity Linking | 2,000 | 1,601 |
| Baidu-FEST (Yu et al., 2023a) | Competition | Multi-scenario | KIE | 1,807 | 377 |
| Newspaper | Self-collect | News Pieces from Newspaper | - | - | 422 |
| Medical instruction | Self-collect | Drug instruction& box | - | - | 235 |
| Total | | | | | 5,071 |

Table 2: Composition of MDCD-VQA dataset. "#" means "the number of".

2023a), Gemini (Anil et al., 2023), PALI-X (Chen et al., 2023), Qwen-VL (Bai et al., 2023b) demonstrate their robust generalization ability on DocVQA tasks. Some works (Wei et al., 2023; Ye et al., 2023; Lu et al., 2024; Liu et al., 2024; Fujitake, 2024; Dong et al., 2024b; Chen et al., 2024) have also been devoted to designing LVLMs specifically for document understanding tasks, such as high-resolution requirements, to address the characteristics of document data. However, all of these works were primarily trained and evaluated on English datasets. A recent study (Shi et al., 2023b) also highlights the challenges faced by GPT-4V in languages other than English. Therefore, the establishment of a robust Chinese benchmark would contribute significantly to the advancement of research on current LLMs/LVLMs.

# 3 MDCD-VQA

In this section, we elaborate on the data construction process and provide statistics and analysis for the proposed MDCD-VQA dataset. More details can be found in the supplementary material.

## 3.1 DATA CONSTRUCTION

**Images:** To build a comprehensive multi-domain dataset, we systematically explored and collected Chinese document data from various sources. In this context, "documents" include traditional office documents, tickets, cards, web pages, and other text-intensive image scenes. Our data collection encompasses three main sources with open copyright: (1) public academic datasets, (2) datasets released alongside public competitions, and (3) private datasets collected by our team. All external data from these sources are publicly available. For each dataset, we selected a varying number of images based on the diversity of data formats. Specifically, we computed the distribution and similarity of the image features in each data source and sampled them diversely within the feature space to ensure the final dataset includes as many different styles as possible. For samples with very similar layouts, the number of samples was relatively small. In total, we collected 5,071 document images from 12 data sources, as shown in Table 2. Each image in the dataset is tagged with a document category label to facilitate problem analysis and indexing of different data types. Some category information is taken directly from the dataset itself, while for others, such as HUST-CELL, we manually assigned labels to categorize the data into predefined document categories. Overall, we defined 39 document types across 7 domains (business, culture, education, finance, lifestyle, medical, and transportation) within the dataset. MDCD-VQA is the first multi-domain DocVQA dataset to provide detailed categorization of document types, thereby significantly facilitating research on system adaptation. The specific distribution can be found in the supplementary material.

**Questions and Answers:** When constructing Q&A pairs, we aim to increase question diversity while maintaining alignment with real-world applications. It is important to note that the public datasets used in our dataset originate from sub-domain document understanding tasks, such as KIE and entity linking. These tasks provide high-quality task labels that match the primary application

---

[2]https://developer.huaweicloud.com/develop/aigallery/dataset/detail?id=b81f24ad-aad6-4a3a-b168-bd92c107a3ea

[3]https://github.com/buptlihang/CDLA

requirements in these scenarios. Consequently, we use a combination of semi-automated generation (generation followed by verification) and manual annotation to construct Q&A pairs.

Specifically, we first create more than 100 templates for different task types and field categories, allowing us to generate a wide range of questions. For example, if we have a KIE annotation that labels "[total value: 6.00]" for an invoice sample, we can create an extractive question such as "What is the {total value}?". In addition to the provided information, we generate questions based on other automatically obtained details such as type, size, and position. To improve the diversity of questions, we also use natural language processing (NLP) augmentation techniques, such as inter-translation and synonym replacement, to further augment the questions. We randomly select 2-8 questions for each image based on the amount of label information originally provided.

Next, we hired 10 Master's level native Chinese annotators and developed a web-based annotation platform. Their tasks included reviewing and modifying the previously automatically generated Q&A pairs and providing additional annotations for each image. During this phase, annotators were encouraged to submit more complex questions, including those that require judgment, reasoning, summarization, or calculation. They were also encouraged to submit questions that cannot be answered from the image. Each Q&A pair was double-checked by at least one other annotator. The final MDCD-VQA dataset consists of 34,170 Q&A pairs. Approximately 50% of the dataset was created through semi-automated generation, and the rest was created entirely by hand.

## 3.2 STATISTICS AND ANALYSIS

Here we show some statistics and analysis of the MDCD-VQA dataset. More information can be found in the Appendix.

| Dataset | Document Tokens | Questions Tokens | Answers Tokens |
|---|---|---|---|
| DocVQA | $183.0 \pm 150.0$ | $8.3 \pm 3.0$ | $2.1 \pm 1.7$ |
| VisualMRC | $154.2 \pm 79.3$ | $9.4 \pm 4.0$ | $8.4 \pm 6.4$ |
| InfographicsVQA | $288.0 \pm 214.6$ | $11.6 \pm 3.7$ | $1.7 \pm 1.4$ |
| DuReader$_{vis}$ | $1,986.2 \pm 1,211.1$ | $10.4 \pm 3.2$ | $180.5 \pm 309.2$ |
| MDCD-VQA | $415.2 \pm 444.3$ | $11.1 \pm 6.2$ | $8.7 \pm 15.1$ |

Table 3: Token length ($Avg \pm std$) comparisons with some single-page-based datasets.

**Tokens length:** Table 3 shows the token lengths of the document text, questions, and answers in the dataset, along with a comparison to some previous datasets. The MDCD-VQA dataset features a rich distribution of text, question, and answer lengths. The DuReader$_{vis}$ dataset contains very long document text and answer lengths, primarily because it consists of high-resolution screenshots of web pages, with question-answer pairs derived from extensive segments searched by Internet engines. Our dataset includes both text-intensive documents (*e.g.,* newspapers, academic papers) and less text-intensive documents (*e.g.,* invoices, business cards), reflecting a variety of scenarios. This diversity supports a more comprehensive evaluation of different methods in current research.

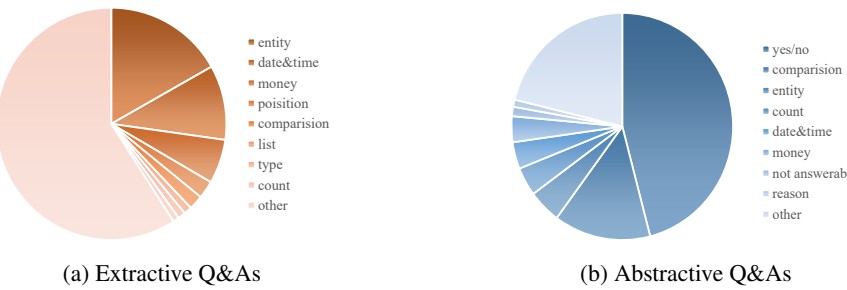

(a) Extractive Q&As                    (b) Abstractive Q&As

Figure 2: The distribution of the frequent Q&As in MDCD-VQA.

**Q&A types**: Like most previous works, we classify Q&A pairs into two types: extractive and abstractive. In the MDCD-VQA dataset, approximately 68% of the questions are extractive, while 32% are abstractive. A question is considered extractive if the answer consists of text present in the

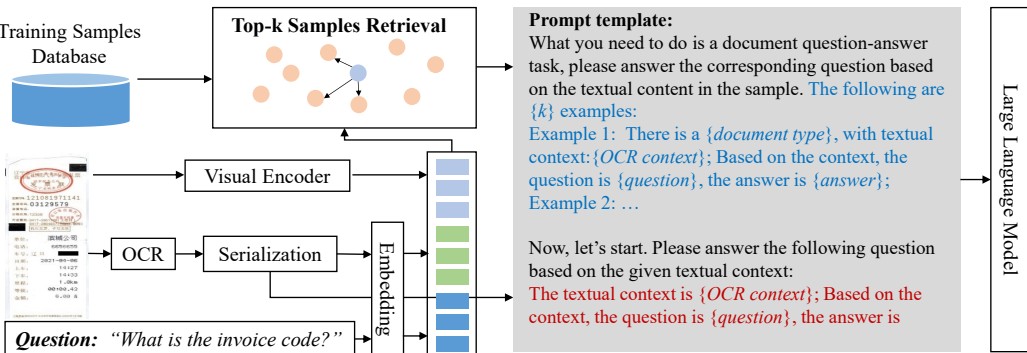

Figure 3: The overall framework of the proposed method. The model first retrieves the top-k nearest samples from training databases and from visual, text and question perspectives. Then, the model uses their OCR context and result to construct the few-shot prompt. The prompt is translated into English for better understanding.

image. In addition to basic information extraction, there are also many complex extractive questions, including those involving comparisons or lists. Figure 2 shows the approximate distribution of Q&A attributes in the MDCD-VQA dataset. Apart from the *other* type, the most common extractive questions are *entity*-related (*e.g.*, names, locations, companies), while the most common abstractive questions are in a *yes/no* format.

## 4 PROPOSED MODEL

In this section, we present a baseline method based on a simple yet powerful ICL framework (Dong et al., 2023) that leverages the robust capabilities of LLMs to achieve high-quality few-shot DocVQA tasks. The overall framework, illustrated in Figure 3, can be broken down into two key steps: *Nearest Sample Retrieval* (NSR) and *Prompt Construction*.

**Nearest Sample Retrieval:** In context learning with LLMs, training samples typically aren't used directly to update model parameters. Instead, they serve as context input to the LLM, activating its potential capabilities. Different examples presented to the model will elicit different responses from the LLM. Inspired by the idea of Retrieval-augmented Generation (RAG)(Lewis et al., 2020), a straightforward approach is to assign each image the most similar example from the training samples. To measure the similarity between samples, we consider three different feature dimensions: the visual feature, the text within the image, and the question itself. Specifically, for an inference sample, we first use a vision transformer (Dosovitskiy et al., 2021) to extract its visual feature, denoted as $V$, which is then flattened into a one-dimensional feature vector. For the text in the image, we use an offline OCR engine to extract all text instances along with their positions. All texts are concatenated into a sequence according to their bounding boxes using a heuristic method from Qiao et al. (2023). Both the serialized text sequence and the question are then encoded using the BGE-large embedding (Xiao et al., 2023) to obtain the sentence embedding, yielding the text feature vector $T$ and the question feature vector $Q$, respectively. For samples in the training set, all features are precomputed offline and stored in a vector database. Consequently, computing the similarity metric between a given sample and any sample ($[V', T', Q']$) in the database becomes straightforward:

$$D = dis(\lambda_1 V + \lambda_2 T + \lambda_3 Q, \lambda_1 V' + \lambda_2 T' + \lambda_3 Q') \tag{1}$$

where $\lambda_1, \lambda_2, \lambda_3$ are parameters used to balance the importance of the features, and we use cosine similarity to compute the feature distances ($dis()$). Finally, we select the $k$ closest examples in the training set. It is worth noting that the samples are stored in the database at the granularity of the question. Once a sample (a question and its corresponding image) is retrieved, all other samples belonging to the same image are skipped to ensure the retrieval of new images.

**Prompt Construction:** Different prompts may influence the output of the LLM to varying degrees. In this baseline model, we use a simple and straightforward prompt design: "Declare the task, give examples, and ask the question," as illustrated in Figure 3. Specifically, the prompt comprises three

key components that need to be filled in the prompt template: 1) Document Type: For the retrieved samples added to the prompt, we include a document-type cue to help the model correlate underlying knowledge with the textual context. For inference, the category is not added here as it is unknown. 2) OCR Context: The OCR result is concatenated into a long sequence using the same ordering strategy and separated by whitespace to distinguish different text instances. 3) Q&A Pairs: Only the retrieved Q&A pairs are concatenated into the prompt, providing format information for the LLM to reference. All the prompts are in Chinese.

## 5 EVALUATION

### 5.1 COMPARED METHODS

As is widely recognized, new large models are being continuously developed and rapidly improving performance metrics. In this study, we have gathered as many advanced models as possible that support Chinese document understanding and utilized their latest versions. Specifically, we compare four types of methods: 1) *Language-Model-based*: BERT (Devlin et al., 2019) and ERNIE3.0 (Liu et al., 2023a); 2) *Multimodal-Pretrained-Model-based*: LayoutXLM (Xu et al., 2021b), LayoutLMv3 (Huang et al., 2022), ERNIE-Layout (Peng et al., 2022); 3) *LLM-based*: InterLM2.5-7B (Zhang et al., 2024), Qwen-14B (Bai et al., 2023a), Qwen2-72B (Yang et al., 2024), ChatGPT (gpt-3.5-turbo) (Ouyang et al., 2022), GPT-4o (OpenAI, 2024) (with **P**ure **T**ext input, gpt-4o-2024-05-13, ); 4) *LVLM-based*: Qwen-VL-7B (Bai et al., 2023b), InternLM-XComposer-2.5 (Dong et al., 2024a), MiniCPN-V2.6 (Yao et al., 2024), CogVLM2 (Hong et al., 2024), InternVL2 (Chen et al., 2024), Qwen-VL-Max (Bai et al., 2023b), Claude 3 Opus (claude-3-opus-20240229) (Anthropic, 2024), Gemini 1.5 Pro (Reid et al., 2024), GPT-4 (Yang et al., 2023a) (gpt-4-turbo) and GPT-4o (with **M**ulti-**M**odal input, gpt-4o-2024-05-13).

For the previous two types of methods, we uniformly adopted the base-scale Chinese version. Following standard practice, we trained the models using the full training set until convergence. For closed-source methods, such as Qwen-VL-MAX, ChatGPT, GPT-4, GPT-4o, Claude-3 Opus, and Gemini 1.5 Pro, we evaluated them through their official APIs. In the few-shot testing, we compared our NSR method with the Random-Sample-Retrieval (RSR) method (Liu et al., 2023b). The proposed few-shot approach can theoretically be applied to both LLM and LVLM models. In this paper, we report only the few-shot testing performance of LVLM-based methods on certain open-source models, as we found similar limitations in most LVLM models' ability to comprehend the examples, including several tests with GPT-4o.

### 5.2 IMPLEMENTATION DETAILS

The MDCD-VQA dataset has been randomly divided into 3,557/ 761/ 753 for training/ validation/ testing, respectively. They separately contain 24,044/ 4,967/ 5,159 Q&A pairs.

Except for the LVLM-based method, the off-the-shell OCR results are obtained using the DavarOCR (Qiao et al., 2022) engine. For the LLMs/LVLMs-based methods, we only evaluate these methods in a zero/few-shot setting, where the training data are used as queryable examples, since tuning a large model is beyond the scope of this work. The weight balance parameters in our NSR method are set as $\lambda_1 = \lambda_2 = \lambda_3 = 1$. All experiments are performed on 8 Tesla A100-80G GPUS.

We adopt the widely used Average Normalized Levenshtein Similarity (ANLS) (Biten et al., 2019) as the primary evaluation metric, which allows partial credit for answers that are close, though not exact matches. Additionally, we follow the approach in Mathew et al. (2021) to report accuracy (Acc), which accounts for exact matches. In Chinese, however, different expression habits may result in answers having the same meaning but entirely different text (*e.g.*, "BúShì" and "MéiYǒu" both convey the meaning of "No"). To address this, we perform simple post-processing to align the model output during evaluation. It is important to note that large models may produce semantically similar responses that are incorrectly judged due to differences in expression. While some methods use large models like GPT for semantic evaluation, they still face significant challenges in achieving accurate assessments. Considering both the evaluation of previous datasets and the need for a stable metric in real-world applications, we continue to use the traditional evaluation method. More details can be found in the Appendix.

| Setting | Types | Models | Params | ANLS | $ANLS_{ex}$ | $ANLS_{abs}$ | Acc | $Acc_{ex}$ | $Acc_{abs}$ |
|---|---|---|---|---|---|---|---|---|---|
| Full-training | Language-Model-based | BERT | 110M | 15.40 | 21.13 | 2.91 | 7.37 | 10.34 | 0.86 |
| | | ERNIE3.0 | 118M | 18.91 | 24.26 | 7.24 | 10.93 | 14.47 | 3.21 |
| | Multimodal-Pretrained-Model-based | LayoutXLM | 352M | 57.61 | 81.01 | 6.54 | 53.54 | 76.99 | 2.34 |
| | | LayoutLMv3 | 266M | 57.19 | 78.76 | 10.12 | 51.85 | 73.97 | 3.58 |
| | | ERNIE-Layout | 277M | 62.92 | 85.82 | 14.32 | 56.76 | 80.89 | 5.57 |
| 0-shot | LLM-based | InternLM2.5-7B | 7B | 67.30 | 75.21 | 50.03 | 60.65 | 69.28 | 41.83 |
| | | Qwen-14B | 14B | 70.34 | 78.71 | 52.08 | 62.88 | 72.39 | 42.13 |
| | | Qwen2-72B | 72B | 75.18 | 80.78 | 63.69 | 70.65 | 76.91 | 57.82 |
| | | ChatGPT | unk. | 71.71 | 81.10 | 51.22 | 65.65 | 76.68 | 41.58 |
| | | GPT-4o (PT) | unk. | 79.11 | 88.43 | 58.92 | 72.77 | 82.98 | 50.66 |
| | LVLM-based | Qwen-VL | 7B | 34.73 | 33.87 | 36.61 | 23.65 | 19.79 | 32.08 |
| | | InternLM-XComposer2.5 | 7B | 64.91 | 73.57 | 45.98 | 53.86 | 61.20 | 37.82 |
| | | MiniCPM-V2.6 | 8B | 66.35 | 72.30 | 53.38 | 58.46 | 64.08 | 46.21 |
| | | CogVLM2 | 19B | 67.96 | 76.65 | 48.98 | 56.21 | 63.62 | 40.04 |
| | | InternVL2-26B | 26B | 71.51 | 79.63 | 53.80 | 60.67 | 67.95 | 44.79 |
| | | Qwen-VL-MAX | unk. | 69.27 | 75.03 | 56.70 | 57.07 | 61.31 | 47.81 |
| | | Claude 3 Opus | unk. | 46.63 | 50.68 | 37.86 | 35.34 | 38.91 | 27.63 |
| | | Gemini 1.5 Pro | unk. | 58.11 | 61.42 | 50.94 | 43.87 | 44.68 | 42.11 |
| | | GPT-4 | unk. | 42.55 | 41.64 | 44.51 | 32.64 | 30.09 | 38.16 |
| | | GPT-4o (MM) | unk. | 70.88 | 78.89 | 53.55 | 58.21 | 64.44 | 44.74 |
| 5-shot | LLM-based | RSR + Qwen-14B | 14B | 67.85 | 76.09 | 49.88 | 60.57 | 69.39 | 41.33 |
| | | RSR + Qwen2-72B | 72B | 79.21 | 86.65 | 63.94 | 75.00 | 83.37 | 57.82 |
| | | RSR + ChatGPT | unk. | 73.22 | 81.49 | 55.42 | 66.10 | 75.72 | 45.38 |
| | | RSR + GPT-4o (PT) | unk. | 75.50 | 83.54 | 58.11 | 70.89 | 80.24 | 50.66 |
| | | NSR + Qwen-14B | 14B | 74.98 | 81.80 | 60.08 | 67.90 | 75.95 | 50.34 |
| | | NSR + Qwen2-72B | 72B | 82.35 | 88.68 | 69.77 | 75.99 | 81.13 | **63.29** |
| | | NSR + ChatGPT | unk. | 79.04 | 86.42 | 62.94 | 72.42 | 81.74 | 52.07 |
| | | NSR + GPT-4o (PT) | unk. | **84.09** | **89.69** | **71.98** | **77.96** | **84.80** | 63.16 |
| | LVLM-based | RSR+Qwen-VL | 7B | 30.77 | 28.49 | 35.75 | 19.89 | 14.67 | 31.28 |
| | | RSR+InternLM-XComposer2.5 | 7B | 52.27 | 58.90 | 37.78 | 41.04 | 46.01 | 30.17 |
| | | RSR+MiniCPM-V2.6 | 8B | 65.97 | 74.22 | 47.96 | 55.30 | 62.44 | 39.73 |
| | | RSR+InternVL2-26B | 26B | 64.69 | 73.70 | 45.03 | 53.50 | 61.84 | 35.29 |
| | | NSR+Qwen-VL | 7B | 32.69 | 29.54 | 39.55 | 22.35 | 17.07 | 33.87 |
| | | NSR+InternLM-XComposer2.5 | 7B | 54.46 | 59.12 | 44.51 | 43.76 | 47.36 | 36.07 |
| | | NSR+MiniCPM-V2.6 | 8B | 73.95 | 80.07 | 60.59 | 63.99 | 69.84 | 51.20 |
| | | NSR+InternVL2-26B | 26B | 65.60 | 71.52 | 52.66 | 55.15 | 60.43 | 43.62 |

Table 4: Summary of performance on the test set of MDCD-VQA dataset. The results with subscripts $ex$ and $abs$ represent the performance on extractive and abstractive questions, respectively. The methods highlighted with shading indicate the best performance in each category.

## 5.3 RESULTS

Table 4 presents the overall evaluation results on the MDCD-VQA dataset.

**In the full-training setting**, ERNIE-Layout demonstrates the highest performance among traditional methods, highlighting its effectiveness in multimodal modeling. However, its proficiency varies across question types, with strong results for extractive questions but weaker performance for abstractive questions, due to the limitations of the task paradigm. It is worth noting that in addition to the indexing modeling paradigm, some generative-based pre-training models also exist, such as UDOP (Tang et al., 2023) and GenDoc (Feng et al., 2023). However, to our knowledge, none of these models provide Chinese pre-training models.

**In the 0-shot setting,** the evaluated models are all effective in understanding Chinese commands. As can be seen from the results, GPT-4o(PT) and Intern-VL-26B have achieved the best results in the LLM-Based and LVLM-based methods, respectively. For the open source models shown so far, their performance is almost positively correlated with the number of parameters. Comparing the two type of schemes, LVLM-based models still lag behind LLM-based methods in performance, as compared between GPT-4o(PT) and GPT-4o(MM).This disparity is primarily due to the fact that LLM-based methods integrate the expertise of OCR specialists, resulting in more accurate text recognition. However, they have an intrinsic flaw: they lose part of the visual information in the processing pipeline, which hampers their ability to handle questions involving visual intricacies.

**In the 5-shot setting**, incorporating examples extracted using our NSR method significantly enhanced the performance of zero-shot LLM-based approaches, improving overall results by 5%-7% and by 8%-13% for abstractive questions. One of the obvious features is that ICL has helped to

constrain the model in the format of the output, thus improving the match with the ground truth. This is actually of great importance for industrial production as well. Among LLM-based methods, NSR+GPT-4o achieved the highest performance across all compared methods. We found that integrating NSR partially mitigates the issue of missing layout information by enabling the model to discern similar patterns from examples. In contrast, the strategy of randomly selecting examples (RSR) provided limited benefit and, in some cases, even underperformed compared to the zero-shot approach. For LVLM models, most NSR/RSR methods showed limited effectiveness, with the exception of MiniCPM-V2.6, particularly for extractive questions. In some cases, these methods led to a significant decrease in performance, with a common error pattern being the models directly replicating answers from retrieved examples instead of using them as references—indicating deficiencies in contextual comprehension (an example is provided in Appendix). This highlights a potential training data gap in the ICL capabilities of other LVLM models.

### 5.4 ABLATION STUDY

In the following experiments, we use the NSR + Qwen-14B (5-shot) setting as a baseline.

**Number of Examples:** First, we experiment with the number of examples concatenated into the prompt. Figure 4 shows the performance when changing the number of examples from 0 to 5 in our framework. There is a clear improvement in the model's performance when we increase the number of examples from 0 to 1 or 2, while after 3 examples, the performance tends to stabilize. This is mainly because the examples are given in order of relevance, and the first example already provides the most similar answer and activates the model's capability accordingly. The more complex the problem, the more examples the model may need to refer to. However, incorporating more examples requires correspondingly larger resource consumption. Therefore, in practical deployment, we need to balance the complexity of the task and the cost of reasoning.

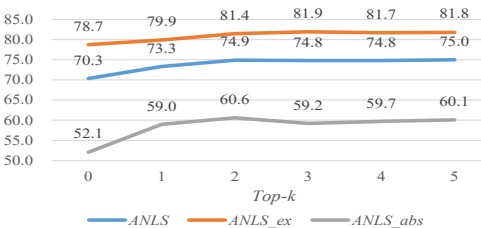

| Features | | | ANLS | $\text{ANLS}_{ex}$ | $\text{ANLS}_{abs}$ |
|---|---|---|---|---|---|
| Image | Text | Question | | | |
| | | | 67.85 | 76.09 | 49.88 |
| ✓ | | | 70.30 | 78.76 | 51.85 |
| | ✓ | | 70.15 | 78.55 | 51.80 |
| | | ✓ | 72.89 | 79.01 | 59.52 |
| ✓ | ✓ | | 70.24 | 78.56 | 52.07 |
| ✓ | | ✓ | 74.84 | 81.30 | 60.76 |
| | ✓ | ✓ | 74.23 | 81.06 | 59.31 |
| ✓ | ✓ | ✓ | 74.98 | 81.80 | 60.08 |

Figure 4: Ablation on the retrieved samples number.     Table 5: Ablation on features used in sample retrieval.

**Features Used in Sample Retrieval**. The proposed method retrieves samples based on three types of features. Table 5 presents the experimental results of ablation studies on the different features utilized. If the model does not employ any of the three features, it defaults to the RSR setting. The results indicate that using even a single feature can enhance the model's performance to some extent. Among the three features, similar question features are the most beneficial, especially for abstractive questions. This is because, for abstractive questions, the answers are not constrained by the text in the image, allowing the model to generate a variety of answer styles. Providing examples of the most similar questions and answers helps the model constrain the style of its answers. When all three features are combined, the model can identify the most similar examples from multiple dimensions, resulting in the highest performance.

## 6 CONCLUSION

This paper introduces a new multi-domain Chinese DocVQA dataset, comprising 39 types of documents from 7 different domains. The dataset includes a diverse set of Q&A pairs, encompassing both extractive and abstractive questions. We conduct a comprehensive comparison of several methods on the Chinese DocVQA task and propose a novel approach based on the in-context learning framework. This approach utilizes image features, text features, and question features to retrieve similar examples from the database, thereby activating the latent capabilities of LLMs/LVLMs. Experimental results demonstrate that our methods establish a new advanced baseline and highlight the strong generalization and few-shot capabilities of the proposed framework.

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

# A APPENDIX

## A.1 ADDITIONAL DATASET STATISTICS

### A.1.1 DOCUMENT CATEGORY

Figure 5 shows the detailed category distribution of the MDCD-VQA dataset. We have divided the documents into 7 domains and 39 categories based on their application scenarios. Note that for many documents that are similar in shape (such as forms), we still categorize them differently according to their usage scenarios.

For some data categories, the amount of data included in the MDCD-VQA is not large enough to fully assess the model's applicability to those domains. However, MDCD-VQA contains several domains (e.g., e-commerce, newspaper, shopping tickets) where the amount of image data is sufficient (comparable to the amount of data in VQA-CD (Mahamoud et al., 2022)) to support domain-specific applicability studies. Unlike most previous datasets, MDCD-VQA focuses more on the general ability and cross-domain generalization of the model, rather than the large size of data in each category.

Although MDCD-VQA covers many scenarios, there are still many categories not represented. In the future, we plan to further expand the coverage of this dataset.

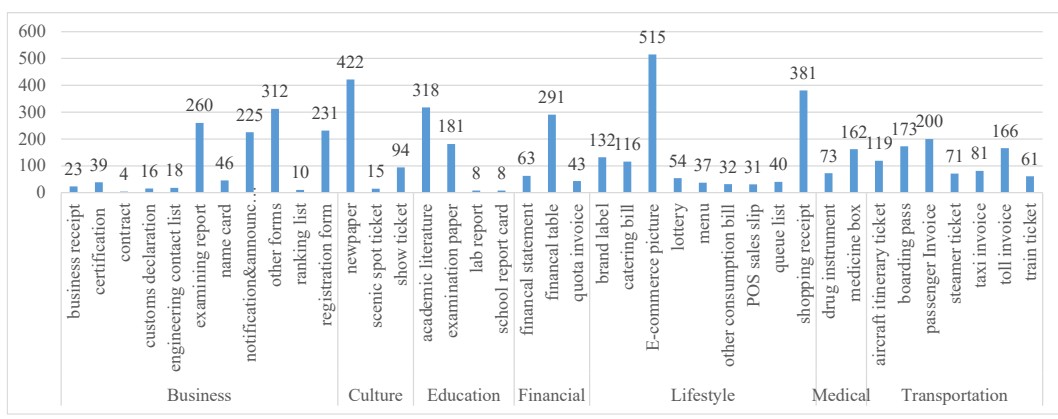

Figure 5: The detailed distribution of the document categories in MDCD-VQA dataset.

### A.1.2 DATASET DIVERSITY

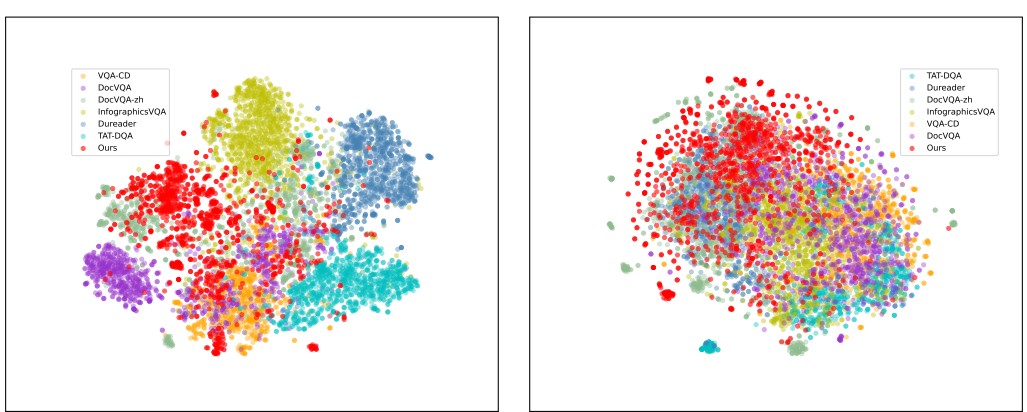

(a) inner image similarity (t-SNE over ResNet50 features of 1k images).

(b) inner text similarity (t-SNE over TF-IDF features of 1k images)

Figure 6: Visualization of inner similarity for different datasets.

There is a rich diversity of images in the MDCD-VQA dataset. Figure 6 illustrates the distribution of visual embeddings (represented by the ResNet50 (He et al., 2016) feature) and textual content embeddings (represented by the BGE embedding (Xiao et al., 2023) feature) of images across different datasets (each dataset randomly selects 1,000 images). This reflects the visual and textual similarities between the samples within a given dataset. From the visual feature distribution, we observe that most previous datasets were concentrated in one or a few clusters, whereas our dataset is distributed across many different clusters. The conclusion for the textual embedding distribution is similar. From the second figure, we can also see that Chinese and English datasets have different distributions, and our data exhibits a relatively scattered distribution within the Chinese domain.

Figure 7 shows the word clouds for text content, questions, and answers according to word frequency in the MDCD-VQA dataset, where only words longer than one character are counted. These word clouds reveal the word frequency distribution in various Chinese documents and Chinese Q&A sentences.

### A.1.3 ANSWER EVIDENCE

Based on the evidence leading to the answers, we categorize the evidence types of question-answer pairs into six categories:

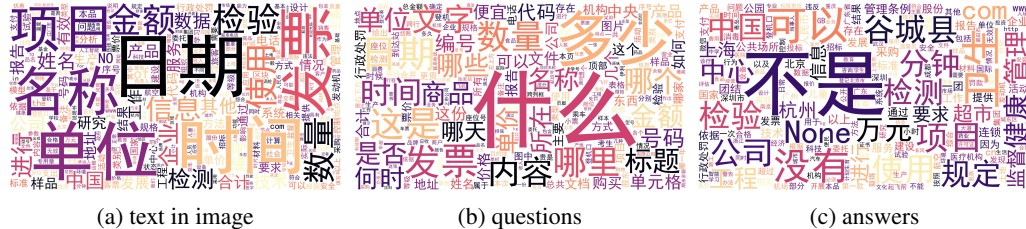

(a) text in image      (b) questions      (c) answers

Figure 7: Word clouds of words in the (a) text in images (b) questions and (c) answers of the MDCD-VQA.

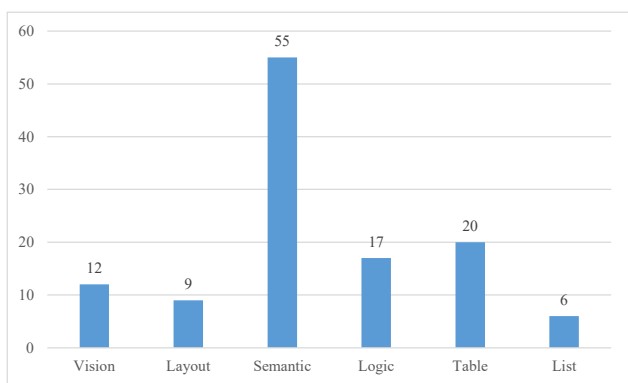

Figure 8: The evidence distribution for 100 samples in test set of MDCD-VQA dataset.

- Visual-related: questions about color, shape, size, stamp, etc.
- Layout-related: questions about position, order, single/double columns, etc.
- Semantic-related: questions about semantic entities, types, etc.
- Logic-related: questions about comparison, calculation, summation, etc.
- Table-related: questions based on tables.
- List-related: questions based on lists.

Note that some questions may belong to more than one type. We randomly selected 100 samples for the test dataset and manually categorized the evidence types. Figure 8 provides a detailed description of the evidence distribution, demonstrating the dataset's diversity and complexity. The results show that the most common evidence type is semantics-related, which is also the most frequent question type in KIE-like tasks.

## A.2 ADDITION IMPLEMENTATION DETAILS

### A.2.1 PROMPT CONSTRUCTION

Here, we provide some extra details for constructing the prompts for LLMs/LVLMs.

**Prompt construction for few-shot LLM**. Note that some documents have very long OCR contexts, and LLM/LVLM prompt tokens usually have a length limit (*e.g.*, 8K for Qwen-14B). Thus, in practical implementation, for samples where the support context length is insufficient for full example splicing, a $k$-shot prompt construction is actually a max-$k$-shot. This means that a maximum number of examples, but no more than $k$, will be added to the prompt. Algorithm 1 presents a simple method to adaptively select the maximum-top-$k$ examples based on the input. In practice, only about 1.6% of test samples cannot be filled with 5 examples due to their context limit.

In the algorithm, we consider that there may be some samples in the queue that cannot be added to the prompt due to the long context or share the same image, so we will continue to traverse the following adjacent samples, and add them to the prompt if there are any tokens left. Here we selected $k + 5$ samples as candidates at once.

---

**Algorithm 1** maximum-to-$k$ examples selection in prompt constructing for LLM

---

**Input:** The queried sample OCR context $T$, Base prompt $B$, Question $Q$, base examples set $E$, current examples set $S$, maximum $k$ samples to retrieve, total token constraint $L$.
**Output:** final example sample $S$.
  Initialize current length $l = length(T + B + Q)$.
  $p = 0$
  retrieve nearest $(k + 5)$ examples from database
  **for** $i = 1$ to $k + 5$ **do**
    **if** $p < k$ and $E_i$'s image not in $S$ and $length(E_i) + l < L$ **then**
      pop $E_i$ from $E$ and push into $S$
      $p = p + 1$
      $l = l + length(E_i)$
    **end if**
  **end for**
  **return** $S$

---

**LVLM Prompt template:**

Your task is to answer the question based on the given document. If the answer is not available from the image, please output "None". The following are {} examples:
Example 1: There is a {*document type*}, {*image path*}</img>, the question is {*question*}, the answer is {*answer*};
Example 2: …

Now, please answer the following question based on the given document:
There is a {*document type*}, {*image path*}</img>, the question is {*question*}, the answer is

Figure 9: An example of a prompt template for LVLM. Texts are translated into English.

**Prompt construction for few-shot LVLM.** Because LVLM input does not contain OCR text content, there is no concern about exceeding the input token limit. An example of an LVLM prompt template is shown in Figure 9.

It is worth noting that different prompts may affect the performance of the model to some extent due to the different training data used between the models and their own ability to follow Chinese prompts. The main goal of this paper is not to study how to design a better prompt, so a relatively simple prompt template is adopted to be as fair as possible to the models under evaluation.

### A.2.2 POST-PROCESSING FOR EVALUATION

For the outputs of LLM/LVLMs, extraneous text content or shifts in output style can make it difficult to assess accuracy precisely. One approach is to send the output results and ground truth values to another LLM (e.g., ChatGPT) for matching judgment, but this method also struggles to guarantee complete accuracy. Here, to prove the efficiency and consistency of the evaluation, we summarize the output styles of various scenarios according to Chinese language expression habits, unify them through simple post-processing, and calculate quantitative indexes through scripts.

Specifically, in the evaluation process, we design post-processing rules to align answers with the same meaning but different formats:

- *Fixed sentence pattern extraction*: For answers in the format "key is value" or "key : value", we keep the answer after the element "is" or ":". For example, for the question "Who is the

passenger of this train ticket?", the prediction "The passenger is Zhangsan" is considered correct if the ground truth is "Zhangsan".

- *Numeric formatting*: For answers in numeric format, such as total money, we remove other characters and compare only the numeric values. For example, for the question "What is the price of this ticket?", the prediction "22 yuan" is considered correct if the ground truth is "22".

- *Yes/No formatting*: For yes/no type answers, we convert all similar expressions to a fixed format. For example, "ShìDe" → "Shì", "MéiYǒu", "Búshì"→ "Fǒu".

- *Unanswerable questions*: For unanswerable questions, if the model's prediction indicates it cannot provide an answer, we marked it as "None" and consider it correct.

## A.3 ADDITIONAL EXPERIMENTAL RESULTS & ANALYSIS

### A.3.1 PERFORMANCE DISTRIBUTION ANALYSIS

Here, we selected some of the representative methods from each category to provide an in-depth analysis of their performance distribution across different domains and question types: 1) traditional methods: ERNIE-Layout, 2) 0-shot LLM-based methods: Qwen2-72B, GPT-4o(PT), 3) 0-shot LVLM-based methods: InternVL2-26B, GPT-4o(MM), 4) few-shot LLM-based methods: NSR+Qwen2-72B, NSR+GPT-4o(PT), 5) few-shot LVLM-based methods: MiniCPM-V2.6.

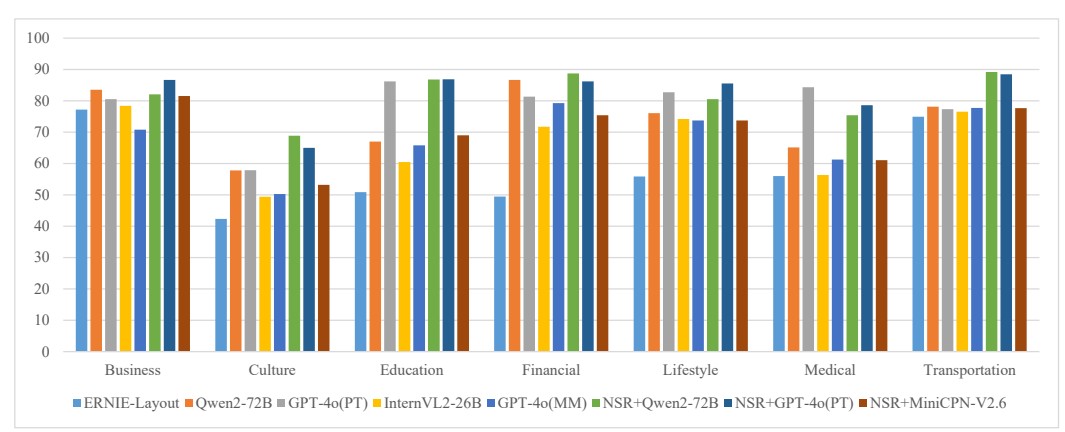

Figure 10: The detailed performance(ANLS) distribution on various document domains.

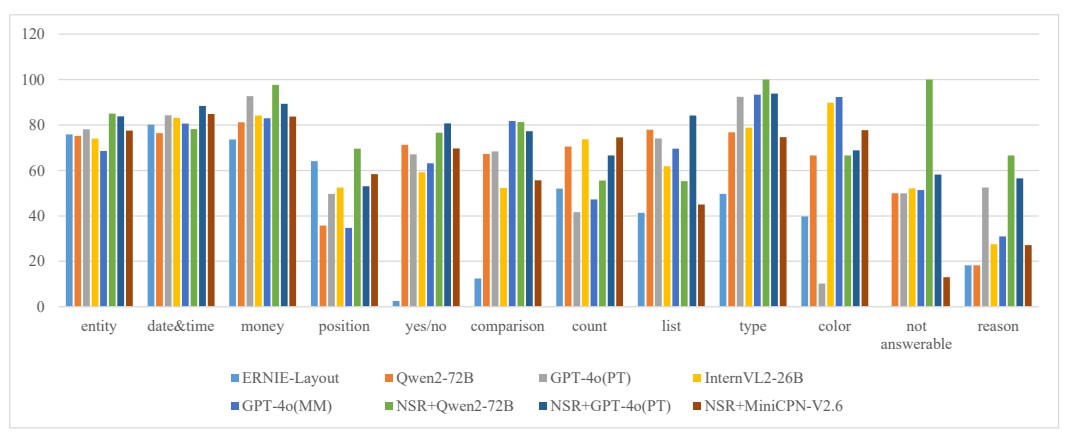

Figure 11: The detailed performance(ANLS) distribution on different question types.

**Performance on different domains:** Figure 10 shows the performance distribution of selected models across seven domains. From the results, we can summarize some characteristics of the different data domains in the MDCD-VQA dataset. For example, in the Business and Transportation domains, where most data consists of office documents or tickets, the percentage of extractive questions is higher. Therefore, ERNIE-Layout achieves relatively high performance. However, this method is limited by its need for substantial training data, making it more suitable for closed-set scenarios. In the Culture domain, the data mainly comprises free-form documents such as newspapers, containing many challenging abstractive questions that require semantic understanding. As a result, the performance of various methods in this domain is relatively low. Additionally, comparing the performance between GPT-4o(PT) and GPT-4o(MM), provides insights into the OCR perception difficulty in this domain. In the financial domain, much of the data is converted from Born Digital sources, allowing the LVLM-based methods to achieve performance close to that of LLM-based methods. In contrast, the Education and Medical domains present more OCR challenges, such as handwriting and text perspective issues, leading to a performance gap between the two types of methods.

**Performance on different question types:** Figure 11 shows the distribution of evaluation metrics across different question categories. Several interesting conclusions can be drawn from the results presented. First, for more conventional extractive problems such as *entity*, *date&time*, and *money*, the compared methods perform well. However, performance differences are more significant for other problem types. For example, for *position*-related questions, the LLM/LVLM-based models fail to perform well, likely because the definition of location is relatively subjective and occurs infrequently in the current corpora of LLMs/LVLMs. In contrast, ERNIE-Layout can fit such distributions through full training. Additionally, ERNIE-Layout is much less effective on some abstractive questions such as *yes/no*, *comparison*, and *type*. For *color*-related questions, the LVLM-based models outperform the LLM-based methods. This is because LLM-based models essentially lose visual information during the inference process. However, with the addition of NSR, samples with similar problems and layouts allow the model to reasonably infer visual information that would otherwise be unavailable, leading to some performance improvement.

### A.3.2 INFERENCE SPEED OF NSR

The evaluation of time consumption is crucial for real-world deployment scenarios, as it provides a more comprehensive understanding of performance. In Table 6, we compare the inference speed of the Qwen-14B and InternVL2-26B models. The increase in time consumption for NSR results primarily stems from two factors: the sample retrieval process and the longer context length. The difference in time consumption between RSR (where retrieval time is negligible) and NSR can be used to estimate the retrieval time. For LVLM-based methods, more input examples lead to additional image I/O operations, further increasing time consumption.

| Setting | Types | Models | FPS( $avg \pm std$ ) |
|---|---|---|---|
| 0-shot | LMM-based | Qwen-14B | $1.27 \pm 0.72$ |
| | LVLM-based | InternVL-26B | $0.88 \pm 0.85$ |
| 5-shot-RSR | LMM-based | RSR+Qwen-14B | $0.85 \pm 0.69$ |
| | LVLM-based | RSR-InternVL2-26B | $0.19 \pm 0.91$ |
| 5-shot-NSR | LMM-based | NSR+Qwen-14B | $0.47 \pm 0.89$ |
| | LVLM-based | NSR+InternVL2-26B | $0.17 \pm 0.67$ |

Table 6: The inference speed comparison of Qwen-14B and InternVL2-26B.

### A.3.3 VISUALIZATION ANALYSIS

We illustrate some visualization results for different methods in Figures 12, 13 and 14. These examples highlight the distinct output characteristics of various models.

Figure 12 compares model responses to extractive questions. While the majority of methods produce similar answers, differences arise in the format of the responses. For example, some models may predict additional irrelevant text, which can be attributed either to errors in model recall or to the model's output style. Although this issue does not indicate a flaw in the models themselves, in practical deployment, we prefer models to produce output in a predefined and fixed format rather

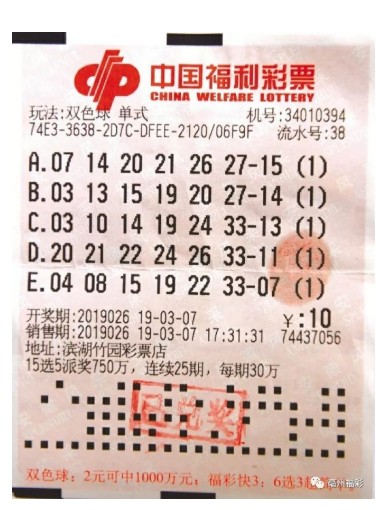

**Question:** 购买的第二组号码是什么? *(What is the second purchased set of numbers?)*

**GT:** *03 13 15 10 20 27-14*

| Full-training | LayoutLMv3 | 1993/7/1 | ERNIE-Layout | 74437056 |
|---|---|---|---|---|
| 0-shot LLM | Qwen-14B | 03 13 15 1920 27-14 | Qwen2-72B | 03 13 15 19 20 27-14 |
| | ChatGPT | 03 13 15 19 20 27 | GPT-4o(PT) | 03 13 15 19 20 27-14 |
| 0-shot LVLM | InternVL2-26B | B.03 13 15 19 20 27-14 (1) | Qwen-VL-Max | 03 13 15 19 20 27-14 |
| | GPT-4V | 购买的第二组号码是：03 13 15 19 20 27-14 (1) | GPT-4o(MM) | 购买的第二组号码是：03 13 15 19 20 27-14 (1) |
| | Gemini-1.5-Pro | 03 10 14 19 24 33-13 | Claude-3-opus | 03 13 15 19 20 27-14 (1) |
| 5-shot LLM | NSR+Qwen-14B | 03 13 15 19 20 27-14 | NSR+Qwen2-72B | 03 13 15 19 20 27-14 |
| | NSR+ChatGPT | 03 13 15 19 20 27-14 | NSR+GPT-4o(PT) | 03 13 15 19 20 27-14 |

**Question:** 具体买了哪些东西? *(What specifically did the shopper buy?)*

**GT:** *晶心低钠盐400g, 水溶c100*

| Full-training | LayoutLMv3 | 晶心低钠盐400g 水溶c100 | ERNIE-Layout | 晶心低钠盐400g |
|---|---|---|---|---|
| 0-shot LLM | Qwen-14B | 晶心低钠盐400g、水溶c100 | Qwen2-72B | 买了晶心低钠盐400g 2个，水溶c100 1个。 |
| | ChatGPT | 晶心低钠盐400g, 水溶c100 | GPT-4o(PT) | 晶心低钠盐400g, 水溶c100 |
| 0-shot LVLM | InternVL2-26B | 晶心低钠盐400g 和水溶c100 | Qwen-VL-Max | None |
| | GPT-4V | 买了哪些东西: 1. 钙心便携瓶装 400g 2. 水蓝 C100 | GPT-4o(MM) | 韧心低钠盐400g 两包，水溶 C100一瓶。 |
| | Gemini-1.5-Pro | 买了2包晶心低钠盐400g和1瓶水溶。 | Claude-3-opus | 酪心低钠盐400g, 水溶c100。 |
| 5-shot LLM | NSR+Qwen-14B | 晶心低钠盐400g、水溶c100 | NSR+Qwen2-72B | 晶心低钠盐400g, 水溶c100 |
| | NSR+ChatGPT | 晶心低钠盐400g、水溶c100 | NSR+GPT-4o(PT) | 晶心低钠盐400g、水溶c100 |

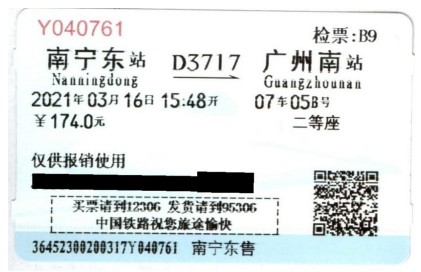

**Question:** 红色文字是什么? *(What is the content of the red text?)*

**GT:** *Y040761*

| Full-training | LayoutLMv3 | Y040761 | ERNIE-Layout | Y040761 |
|---|---|---|---|---|
| 0-shot LLM | Qwen-14B | B9 | Qwen2-72B | 在给出的文字内容中并没有提到任何红色文字，因此无法从给定信息中得出答案 |
| | ChatGPT | 南宁东站 | GPT-4o(PT) | 南宁东站 |
| 0-shot LVLM | InternVL-2-26B | Y040761 | Qwen-VL-Max | Y040761 |
| | GPT-4V | 红色文字是 "Y040761"。 | GPT-4o(MM) | Y040761 |
| | Gemini-1.5-Pro | None | Claude-3-opus | 发货报销使用 |
| 5-shot LLM | NSR+Qwen-14B | Y040761 | NSR+Qwen2-72B | Y040761 |
| | NSR+ChatGPT | Y040761 | NSR+GPT-4o(PT) | Y040761 |

Figure 12: Some visualization results for different methods.

**Question:** 文字"22,182,851.07"的框是跨列框吗？ (Is the text "22,182,851.07" in a spanning cell?)

GT: 不是(no)

| Full-training | LayoutLMv3 | 所得税费用 | ERNIE-Layout | 22 |
|---|---|---|---|---|
| 0-shot LLM | Qwen-14B | 是 | Qwen2-72B | None |
| | ChatGPT | 是 | GPT-4o(PT) | None |
| 0-shot LVLM | InternVL-2-26B | 是 | Qwen-VL-Max | 是 |
| | GPT-4V | 不是 | GPT-4o(MM) | 是 |
| | Gemini-1.5-Pro | 是 | Claude-3-opus | 是 |
| 5-shot LLM | NSR+Qwen-14B | 不是 | NSR+Qwen2-72B | 不是 |
| | NSR+ChatGPT | 不是 | NSR+GPT-4o(PT) | 不是 |

**Question:** 这份文档是单栏吗？ (Is this document a single-column document?)

GT: 是 (yes)

| Full-training | LayoutLMv3 | 规模 | ERNIE-Layout | 抓 |
|---|---|---|---|---|
| 0-shot LLM | Qwen-14B | None | Qwen1.5-72B | None |
| | ChatGPT | 是的。 | GPT-4o(PT) | None |
| 0-shot LVLM | InternVL-2-26B | | Qwen-VL-Max | 是 |
| | GPT-4V | 是 | GPT-4o(MM) | 是的，这份文档是单栏的。 |
| | Gemini-1.5-Pro | 是。 | Claude-3-opus | 根据图片显示，这份文档的版式采用的是双栏排版，而不是单栏。因此问题"这份文档是单栏的吗？"的答案是：No |
| 5-shot LLM | NSR+Qwen-14B | 是 | NSR+Qwen2-72B | 是 |
| | NSR+ChatGPT | 是 | NSR+GPT-4o(PT) | 是 |

**Question:** 左下方的文字内容是什么？ (What is the text in bottom-left?)

GT: [不良反应] 尚不明确

| Full-training | LayoutLMv3 | [不良反应]尚不明确 | ERNIE-Layout | [不良反应]尚不明确 |
|---|---|---|---|---|
| 0-shot LLM | Qwen-14B | 本品为深褐色的水丸；气芳香，味微苦。 | Qwen2-72B | 左下方的文字内容是药品说明书的信息，包括药品名称、警示语、成份、性状、功能主治、规格、用法用量和不良反应等。但是，具体到"左下方"这个描述在给出的文字中并不适用，因为这里并没有分区域描述。所以，更准确的回答应该是这段文字整体的内容描述，而不宜仅指"左下方"。如果必须回答，可以考虑回答为："脑立清丸说明书的部分内容"，但这并不是一个非常精确的答案。因此，我的回答是 \"None\"，因为原问题中的"左下方"在给定文本中没有具体对应。 |
| | ChatGPT | None | GPT-4o(PT) | OTC请仔细阅读说明书并按说明使用或在医师指导下购买和使用。警示语：孕妇及体弱虚寒者忌服；本品含清半夏 |
| 0-shot LVLM | InternVL2-26B | None | Qwen-VL-Max | 葵花 脑立清丸说明书 OTC |
| | GPT-4V | 左下方的文字内容是："不良反应"、"可不明确" | GPT-4o(MM) | 规格：每10丸重1克。 |
| | Gemini-1.5-Pro | 尚不明确。 | Claude-3-opus | 尚不明确。 |
| 5-shot LLM | NSR+Qwen-14B | 孕妇及体弱虚寒者忌服 | NSR+Qwen2-72B | 由于提供的样本描述中并没有提及左下方的具体文字内容，我们只能依据给定的信息来回答问题。在描述中，最后提到的是药品的用法用量："口服。一次10丸，一日2次。"因此，如果假设问题中的"左下方的文字内容"指的是样本描述的最后一部分信息，那么答案应该是关于药品用法用量的部分。但是，严格来说，问题要求的是左下方的文字内容，而描述中并未具体指出这部分内容，所以无法直接从给定信息中准确回答这个问题。\n\n然而，为了遵循问题的回答格式，我们可以推测性地给出一个答案，即药品的用法用量，但这并不是基于明确指示的左下方内容。因此，答案是："口服。一次10丸，一日2次。"但请注意，这仅是一种基于常见药品说明书布局的推测，并非直接来源于问题描述中的明确信息。 |
| | NSR+ChatGPT | {不良反应}尚不明确 | NSR+GPT-4o(PT) | 左下方的文字内容是："孕妇及体弱虚寒者忌服；本品含清半夏" |

Figure 13: Some visualization results for different methods.

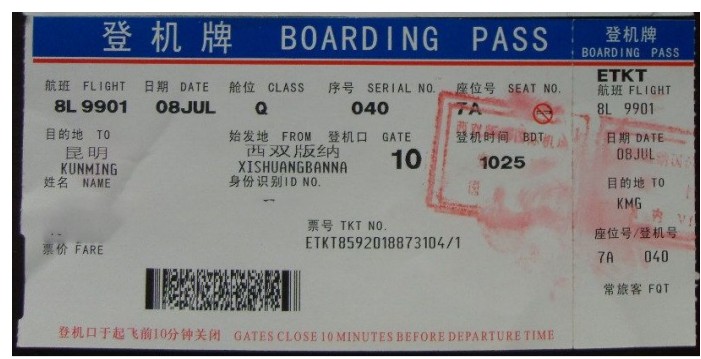

**Question:** *最晚什么时间到达登机口？ (What is the latest time to arrive at the gate?)*

**GT:** *10:15*

| Full-training | LayoutLMv3 | 1025 | ERNIE-Layout | 10 |
|---|---|---|---|---|
| 0-shot LLM | Qwen-14B | 登机口于起飞前10分钟关闭。 | Qwen2-72B | 1015 |
| | ChatGPT | 最晚在起飞前10分钟到达登机口。 | GPT-4o(PT) | 起飞前10分钟 |
| 0-shot LVLM | InternVL2-26B | 10:25 | Qwen-VL-Max | 10分钟前 |
| | GPT-4V | 登机口在起飞前10分钟关闭。 | GPT-4o(MM) | 最晚10点15分到达登机口。 |
| | Gemini-1.5-Pro | 起飞前10分钟 | Claude-3-opus | 1025 |
| 5-shot LLM | NSR+Qwen-14B | 10:10 | NSR+Qwen2-72B | 10:15 |
| | NSR+ChatGPT | 10:15 | NSR+GPT-4o(PT) | 10:15 |

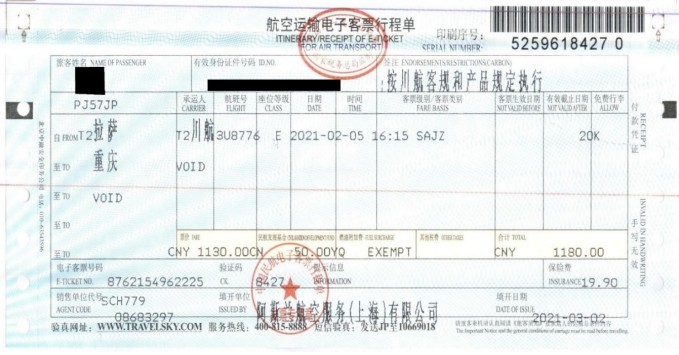

**Question:** *保险费占总票价的百分之几（结果保留两位小数）？ (What percent of the total fare is the insurance premium (retain two decimal places for results)？)*

**GT:** *1.69%*

| Full-training | LayoutLMv3 | 19.9 | ERNIE-Layout | 19 |
|---|---|---|---|---|
| 0-shot LLM | Qwen-14B | 保险费占总票价的百分之几（结果保留两位小数）？ | Qwen2-72B | 虽然可以从文本中找到保险费（￥19.90）和总价（￥1180.00），但是计算百分比需要进行数学运算，这超出了仅从文本中提取信息的要求。因此，根据指示，当对答案没把握时，应回答None |
| | ChatGPT | 0.72% | GPT-4o(PT) | 1.69 |
| 0-shot LVLM | InternVL2-26B | 1.70 | Qwen-VL-Max | 0.01 |
| | GPT-4V | 保险费为19.90元，总票价为1180.00元。保险费占总票价的百分比为：\[ \left( \frac{19.90}{1180.00} \right) \times 100 = 1.68644\% \] 结果保留两位小数，为1.69%。 | GPT-4o(MM) | 保险费占总票价的百分比为：\n\n(19.90 / 1180.00) * 100 = 1.69%\n\n答案是：1.69%。 |
| | Gemini-1.5-Pro | 1.08 | Claude-3-opus | 8.42 |
| 5-shot LLM | NSR+Qwen-14B | 保险费占总票价的百分之几（结果保留两位小数）？ | NSR+Qwen2-72B | 为了计算保险费占总票价的百分比，我们首先需要确定保险费和总票价的数值。从给定的信息中，我们可以看到保险费是￥19.90，总票价是￥1180.00。\n\n计算百分比的公式是：\n\\[ \\text{百分比} = \\left( \\frac{\\text{部分数值}}{\\text{整体数值}} \\right) \\times 100\\% \\]\n\n将具体数值代入公式，我们有：\n\\[ \\text{百分比} = \\left( \\frac{19.90}{1180.00} \\right) \\times 100\\% \\]\n\n计算这个表达式的结果：\n\\[ \\text{百分比} = \\left( \\frac{19.90}{1180.00} \\right) \\times 100\\% = 1.6864406779661016\\% \\]\n\n保留两位小数，结果是：\n\\[ \\text{百分比} = 1.69\\% \\]\n\n因此，保险费占总票价的百分比是1.69%。所以，答案是：1.69%。 |
| | NSR+ChatGPT | 1.68% | NSR+GPT-4o(PT) | 1.75% |

Figure 14: Some visualization results for different methods.

than a verbose one, to facilitate easier downstream use. Notably, in the third subfigure, which presents a question about color, Qwen2-72B correctly indicates that the answer cannot be judged based on the available information, whereas the NSR+Qwen2-72B scenario uses other examples to provide an accurate judgment.

Figure 14 shows examples of more complex problems involving layout analysis, where the differences between models become more pronounced. Since layout information is strongly related to vision, LLM-based approaches do not perform well on these types of questions. Furthermore, ambiguity often exists in understanding layouts, so methods like LayoutLMv3 and ERNIE-Layout, which are fully trained and explicitly model layout modalities, perform better on these tasks.

Figure 15 presents examples that require calculation and reasoning. From the illustration, we observe the powerful reasoning capabilities of many current LLMs and LVLMs. For instance, Qwen2-72B tends to explicitly output part of the reasoning process to ensure the accuracy of the entire computation, although post-processing or special prompt design is still needed for production use.

### A.3.4 FAILURE CASES ANALYSIS

n this section, we provide a summary of the typical types of errors encountered in some scenarios based on large models.

**Errors in 0-shot/few-shot LLM-based methods.** As mentioned earlier, such methods are based on a two-stage pipeline, which inevitably leads to some error accumulation throughout the process. However, the current perceptual results rely on OCR expert realization, making it the solution with the least loss of accuracy at the perceptual level. In summary, the most significant problems stem mainly from the following:

- *Lack of visual information*. Some problems involving color, position, size, and type have inherent flaws due to the absence of visual information. However, in few-shot settings, these issues can be somewhat mitigated. For instance, the model can infer hints from examples containing similar types of problems.

- *Wrong semantic order*. The current heuristic rule for text concatenation can lead to errors, especially in complex layouts like multi-column formats. The reading order of the text may affect the extraction accuracy of the model. This issue could potentially be resolved by introducing a more generalized reading order prediction module (Wang et al., 2021b).

- *Deficiencies in Reasoning Ability:* For tasks involving reasoning and computation, the performance of different large language models may vary significantly due to their inherent capabilities. Some models may even produce many hallucinations in their outputs.

**Errors in 0-shot LVLM-based methods.** This type of scheme is currently more of a black-box approach, making it relatively difficult to pinpoint the root cause of its errors. However, certain types of errors can be identified through more obvious examples. For instance, in the second example shown in Figure 12, it is evident that some of the LVLMs correctly find the position of the answers, but the output textual content is incorrect. This type of problem can be attributed to defects in their perceptual capabilities.

Moreover, while the LVLM-based approach theoretically reduces the accumulation of errors, there are still issues due to the distribution of the large model training corpus. These issues can lead to defects in visually related or inference-related abilities.

**Errors in few-shot LVLM-based methods.** In addition to the previously mentioned issues, we identified a prevalent flaw in approaches involving LVLMs. Specifically, when these models generate answers based on provided examples, they often directly output the answers from the examples rather than performing recognition on the queried images. This observation highlights a significant limitation in the contextual capabilities of LVLMs, likely due to the insufficient inclusion of image-text interleaved data in their training samples.

For instance, consider the NSR+InternVL2-26B model, as shown in Figure 15. When asked, "What is the mode of transportation?" for the queried image, the correct answer should be "river and sea transportation." However, the nearest examples retrieved by NSR all contain the ground truth of "waterway transport." After incorporating these examples into the prompt, the model erroneously extracts the answer "waterway transport" from the prompt and outputs it directly.

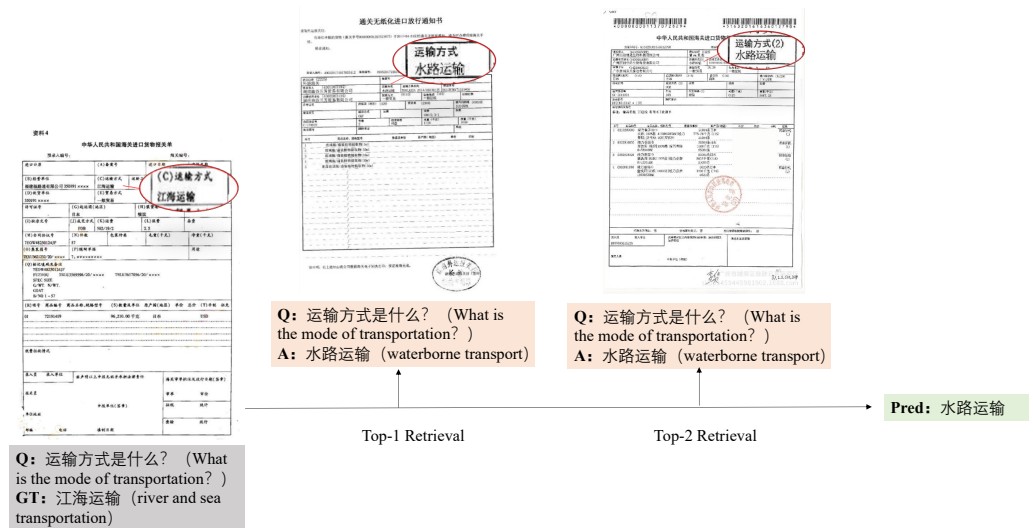

Figure 15: An example of an error case in the NSR+InternVL2-26B method. The model tends to use the answer directly in examples rather than obtaining information from the inferred image.

This issue is not unique to NSR+InternVL2-26B; similar problems were observed across various LVLMs tested. This indicates a broader deficiency in the models' ability to distinguish between example-based suggestions and the actual content of the queried images. Among the many models tested, only MiniCPM-v2.6 was able to use examples to answer some questions correctly, although this was largely due to formatting constraints on the output. Further reviews are needed to explore more ICL capabilities in the future.

A.4    LIMITATION

This work still has some limitations:

- The dataset has not yet been evaluated by a third party to assess human performance. Incorporating third-party evaluations could provide more robust benchmarks and highlight areas for further improvement in both the dataset and the models tested.

- The size of the dataset can be further improved. The current Chinese dataset is relatively small within single domains due to the multi-domain nature of the dataset. The data selection strategy restricts the sampling of data with the same format, and many real-world sample types are not yet covered. We plan to continuously collect and expand the dataset to ensure wider coverage across various domains.

- The tests for the LLMs and LVLMs are not yet comprehensive. The rapid iteration and emergence of new large models in the market mean that some test conclusions may change quickly. Despite this limitation, the proposed dataset can provide a good reference benchmark for evaluating the performance of current large models.

By addressing these limitations, we aim to enhance the dataset's utility and the comprehensiveness of model evaluations, providing a more robust benchmark for future research and development in the field.

