# DATASHEET FOR MDCD-VQA

## 1 DATASHEET

Because of the double-blind mechanism, we have deleted some of the questions, which will be added to the subsequent paper as they become publicly available.

### 1.1 MOTIVATION

**For what purpose was the dataset created?**
At a time when large models are rapidly evolving, research efforts are starting to focus more and more on this end-to-end document understanding capability, and a lot of work has been done to validate their performance on DocVQA tasks. Currently, however, almost all of the work has only demonstrated the effect on English data, which limits the application of related models to other languages, such as Chinese. Moreover, these datasets are usually derived from a single domain, which does not provide a good proof of the generalization ability of the models. We propose this dataset, which on the one hand can help to have a good evaluation of the existing models on the Chinese document understanding task, and on the other hand, the training data and labels it provides can be treated as high-quality data to be added to the training of the large models.

### 1.2 COMPOSITION

**What do the instances that comprise the dataset represent (e.g., documents, photos, people, countries)?**
The MDCD-VQA dataset comprises document images, accompanied by their respective annotation files. These images represent scanned, photographed or Born-Digital-file transferred documents and are stored in the Joint Photographic Experts Group (JPEG) format.

**How many instances are there in total (of each type, if appropriate)?**
The MDCD-VQA contains 5,071 images in total, and a detailed category distribution can be found in paper appendix.

**Does the dataset contain all possible instances or is it a sample (not necessarily random) of instances from a larger set?**
Part of the images of MDCD-VQA are randomly sampled from several public datasets, including EPHOIE, EATEN, SCID, CER-VIR,ComFinTab, CDLA, DI, XFUNSD, HUST-CELL, and Baidu-FEST. For other self-collected samples, we collect them from some public newspaper resources database and searching enging (*e.g.*, Google). The number of samples is based on the diversity of samples in the original dataset. We have annotated these images with VQA labels and category labels.

**What data does each instance consist of?**
Each instance in the dataset consists of an image and associated annotations. These annotations, stored in JSON format, include a list of question-answer pairs, the OCR annotations (bounding boxes and transcriptions for each line of text), and the document category label.

**Is there a label or target associated with each instance?**
Yes.

**Is any information missing from individual instances?**
No.

**Are relationships between individual instances made explicit (e.g., users' movie ratings, social network links)?**

MDCD-VQA desensitizes all information that may contain personally identifiable information (such as a person's name in a ticket).

**Are there recommended data splits (e.g., training, development/validation, testing)?**
The dataset provides a division of the training, validation and test sets, each containing 3,561/ 762/ 755 of images, respectively.

**Are there any errors, sources of noise, or redundancies in the dataset? If so, please provide a description.** There could have some potential noise of QA annotation and OCR annotation.

**Is the dataset self-contained, or does it link to or otherwise rely on external resources (e.g., websites, tweets, other datasets)?**
Some of the data in MDCD-VQA are sampled from other public datasets, and for some of the data where redistribution is restricted, we provide the link to download the original data collection and the indexes of the corresponding sampled data in the data description. All data labels are self-contained.

**Does the dataset contain data that might be considered confidential (e.g., data that is protected by legal privilege or by doctor–patient confidentiality, data that includes the content of individuals' non-public communications)?**
No. All samples in MDCD-VQA are public available.

**Does the dataset contain data that, if viewed directly, might be offensive, insulting, threatening, or might otherwise cause anxiety? If so, please describe why.**
No.

**Does the dataset relate to people? If not, you may skip the remaining questions in this section.**
The vast majority of the data in MDCD-VQA comes from other publicly available datasets, which we have verified have been desensitized to remove personal information when they are made publicly available. Other parts of our own collection do not contain private information directly related to people.

## 1.3 COLLECTION PROCESS

**How was the data associated with each instance acquired?**
The collection process is described in Section 3.1 of the main paper. The data is directly observable.

**What mechanisms or procedures were used to collect the data (e.g., hardware apparatuses or sensors, manual human curation, software programs, software APIs)?**
The annotation process is mainly described in Section 3.1 of the main paper. And we specifically developed a web-based tool for data annotation, along with the detailed instructions on the web page, as shown in Figure 1. With this tool, annotators are able to add, modify, verify, or delete the QA pairs for an image.

**If the dataset is a sample from a larger set, what was the sampling strategy (e.g., deterministic, probabilistic with specific sampling probabilities)?**
The sampling process is a manual selection process in which each image is carefully examined by the dataset constructors. The main guidelines for selection are representativeness and diversity, i.e. based on the diversity of data in the original dataset, for data with a certain layout (e.g. train tickets) the sample size is no more than 50 images because the QA questions that can be constructed based on this data are basically similar. In contrast, for data with a more diverse and rich layout, as many different data types as possible are covered.

**Who was involved in the data collection process (e.g., students, crowdworkers, contractors) and how were they compensated (e.g., how much were crowdworkers paid)?**
For the annotators, they are all Chinese native speakers from our lab or partner labs. According to the total working hours and their average salary, the expenditure for the whole annotation is estimated to be $5000.

**Over what timeframe was the data collected?**
Collection of the dataset began in July 2023 and it took an estimated 4 months to complete data collection and labeling.

**Were any ethical review processes conducted (e.g., by an institutional review board)?**
We have conduct an internal ethical review process by the company's legal compliance department.

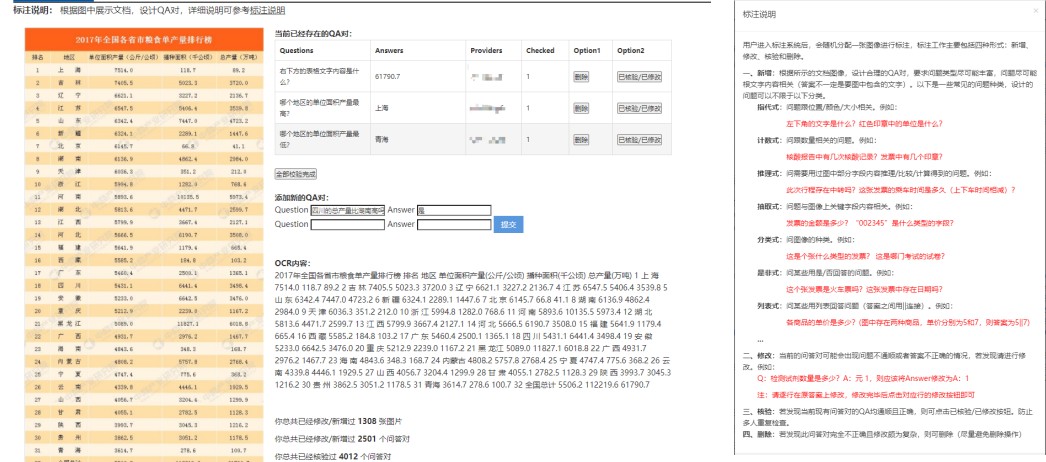

(a) Screenshot of the web-based annotation tool.

(b) Illustration of the instruction for annotators.

Figure 1: We developed a web-based annotation tool, as shown in (a). The annotation tool will randomly push the images to the annotator, who can see the history of the annotated content. In the table on the right side, annotators can directly edit and delete the history of the annotation content, and can also click on the verification button to review a annotation; the annotator can also submit new annotation information in the form below. Below the tool will display the number of images that have been added, modified and reviewed by the current annotator in real time. The annotation instructions is shown in (b).

**Does the dataset relate to people? If not, you may skip the remainder of the questions in this section**

The annotators is only asked to design the QA data based on the images provided, and there is no collection of individual-related information involved.

## 1.4 PREPROCESSING/CLEANING/LABELING

**Was any preprocessing/cleaning/labeling of the data done (e.g., discretization or bucketing, tokenization, part-of-speech tagging, SIFT feature extraction, removal of instances, processing of missing values)?**

The annotations of the MDCD-VQA dataset come from two parts, with each part accounting for about 50%. The first part is based on the original data labels provided by the original public dataset (e.g., the labels of the KIE task), and we randomly selected some of them and transformed them into the form of QA by designing various templates. And the question form is enriched by natural language extensions (e.g., mutual translation). The other part is designed by the annotators directly based on the provided images, and this part of QA will avoid the existing QA style as much as possible to achieve a more liberalized question design.

For each image, annotators can see what has been annotated so far and can choose to modify or delete the current QA content. Each question (including both semi-automated and manually annotated questions) must be reviewed by at least one person (by clicking the Reviewed button). When the number of questions in an image reaches the threshold and all questions have been reviewed, the annotator is no longer pushed. The annotator will continue to annotate or review until no new data is pushed.

**Was the "raw" data saved in addition to the preprocessed/cleaned/labeled data (e.g., to support unanticipated future uses)?**

No. The images contained in the MDCD-VQA dataset have not been additionally processed, and the referenced original data and data labels can be accessed via the provided connection.

**Is the software that was used to preprocess/clean/label the data available?**

Currently not. We will consider open-sourcing this web-based tool in the future.

## 1.5 USES

**Has the dataset been used for any tasks already?**
No.

**Is there a repository that links to any or all papers or systems that use the dataset?**
It is a new dataset that haven't been used by current works. We run existing state-of- the-art models and release the code.

**What (other) tasks could the dataset be used for?** In addition to being used directly for tasks related to large model evaluation or document VQA, this dataset can be used for various OCR or document understanding related tasks, e.g., information extraction tasks can be constructed using only the extractive questions contained in the dataset; document classification tasks can be constructed using the document category labels provided by the dataset; the OCR labels provided in this dataset can be used to improve the generalization ability of the OCR model in various document scenarios, etc.

**Is there anything about the composition of the dataset or the way it was collected and preprocessed/cleaned/labeled that might impact future uses?**
Since the current dataset is new, none of the existing large models have used it in their own training tasks. However, if the dataset is made public, there is a high probability that it will be used by future new big models in their training process (the training process of existing large models generally collects as much high-quality data as possible to improve the generalization ability of the models), which will somewhat affect the fairness of evaluation among models on this dataset.

**Are there tasks for which the dataset should not be used?**
The dataset should not be used for commercial usage.

## 1.6 DISTRIBUTION

**Will the dataset be distributed to third parties outside of the entity (e.g., company, institution, organization) on behalf of which the dataset was created?**
No.

**How will the dataset will be distributed (e.g., tarball on website, API, GitHub)?**
All code and dataset will be publicly distributed on GitHub.

**When will the dataset be distributed?**
It will be published along with the paper.

**Will the dataset be distributed under a copyright or other intellectual property (IP) license, and/or under applicable terms of use (ToU)?**
The dataset is under Creative Attribution-NonCommercial-ShareAlike 4.0 International (CC BY-NC-SA 4.0) License.

**Have any third parties imposed IP-based or other restrictions on the data associated with the instances?**
No.

**Do any export controls or other regulatory restrictions apply to the dataset or to individual instances?**
All authors bear all responsibility for the dataset in case of violation of rights, etc.

## 1.7 MAINTENANCE

**How can the owner/curator/manager of the dataset be contacted (e.g., email address)?**
The contact email will be updated in real time in the introduction page of the dataset.

**Is there an erratum?**
No.

**Will the dataset be updated (e.g., to correct labeling errors, add new instances, delete instances)?**

If significant errors are reported by dataset users, we will consider updating the dataset accordingly. The update information will also be posted on the dataset page.

**If the dataset relates to people, are there applicable limits on the retention of the data associated with the instances (e.g., were the individuals in question told that their data would be retained for a fixed period of time and then deleted)?**
No.

**Will older versions of the dataset continue to be supported/hosted/maintained?**
Yes. If we plan to update the data, we will keep the original version available and then release the follow-up version.

**If others want to extend/augment/build on/contribute to the dataset, is there a mechanism for them to do so?**
Yes, others who are providing some issue fixes can just make a pull request on Github or contact us privately. If it's to expand new work based on our data, it just needs to follow the license.