# OpenReview forum: "Towards Multi-Domain Chinese Document VQA: a New Dataset and Baseline Method"
_ICLR.cc/2025/Conference — ICLR 2025 Conference Withdrawn Submission_

### Official Review · Reviewer_bsVK · 2024-11-01

**Soundness:** 2
**Presentation:** 3
**Contribution:** 2
**Rating:** 3
**Confidence:** 4

**Summary:**

The paper's primary contribution lies in introducing a novel multi-domain Chinese DocVQA dataset encompassing 39 document types across 7 domains, incorporating both extractive and abstractive questions, thereby addressing the gap in Chinese document understanding resources. Additionally, it presents a robust baseline leveraging few-shot in-context learning, demonstrating superior performance across various technical paradigms through comprehensive comparative evaluations.

**Strengths:**

The core contribution of this work is that it introduces a novel Chinese DocVQA dataset. This dataset encompasses multiple domains and addresses a critical gap in non-English document understanding benchmarks. The contributions are commendable additions to the field.

**Weaknesses:**

Despite these contributions, I have several concerns that need to be addressed.
1. The contribution of this work is limited. As shown in Table 2, the proposed dataset primarily derives from existing open-source benchmark datasets.
2. I would like to understand the motivation behind employing in-context learning for the baseline method. The manuscript appears to lack sufficient explanation for this aspect.
3. I suggest expanding the comparative evaluation in Table 4, which currently focuses on general-purpose multimodal large models, to include specialized document-oriented multimodal models such as MplugDoc and Monkey. Poor Chinese language performance from these specialized models would further validate the significance of the proposed dataset.
4. The literature review is inadequate, particularly in its discussion of multimodal large models for document understanding. This gap in the related work section needs to be addressed.
5. With respect to the references, attention should be paid to the correct capitalization of model names (e.g., "Qwen-vl" should be "Qwen-VL", "Layoutllm" should be "LayoutLLM"). A thorough review of all references is recommended to ensure proper formatting.

**Questions:**

See the weaknesses.

---

### Official Review · Reviewer_A72T · 2024-11-01

**Soundness:** 3
**Presentation:** 3
**Contribution:** 2
**Rating:** 5
**Confidence:** 4

**Summary:**

This paper examines the Document Visual Question Answering (VQA) task, with a focus on Chinese Document VQA. The authors have created a new Chinese Document VQA dataset called MDCD-VQA, comprising 39 document types, 7 different domains, 5,071 images, and 34,170 questions. MDCD-VQA includes both extractive and complex abstractive questions, and can be divided into training, validation, and test sets for fine-tuning. Additionally, the paper proposes a baseline method using in-context learning and evaluates the performance of various Large Language Models (LLMs) and Multimodal Large Language Models (MLLMs) on this new dataset.

**Strengths:**

The paper tackles an important area in the field of multilingual large language models, addressing key challenges in this domain, i.e. the lack of high-quality benchmark dataset in the filed of Chinese Document VQA.

 The authors have collected a comprehensive dataset, comprising 39 document types, 7 different domains, 5,071 images, and 34,170 questions, which serves as a strong foundation for their work.

Additionally, they conduct a thorough benchmarking experiment across various models, accompanied by empirical analysis that effectively highlights the current performance of multilingual large language models.

**Weaknesses:**

One limitation of this paper is that the dataset is not publicly available yet, which limits its accessibility and reproducibility for the reviewers.

Additionally, while the evaluation method is very important for such benchmarks, it lacks sufficient detail and presentation, making the results less convincing. Considering the difference between Chinese and English, the evaluation metrics should be carefully designed to deal with the nuances in the model generation. I have read some discussion in the paper but found it not informative. For example, does the simple post-processing (Line 373) suffice in most cases?

The benchmark experiments are also missing some crucial evaluations, such as the use of visual modalities in state-of-the-art commercial models with few-shot scenarios (e.g., GPT-4o with 5-shot), which could offer insight into the current capabilities on this benchmark. Although high costs may pose a challenge, even a subset of these results would illustrate how well current models perform and provide a clearer indication of the benchmark's difficulty.

Lastly, further elaboration on what makes this benchmark challenging and on the performance analysis of different MLLMs would help underscore its significance and utility. For example, is the difficulty from the visual modality? The best performance is almost all from the 5-shot LLM based setting, does this indicate that as long as current models hav suffice textual information about the input image, the model can answer the questions quite well?

**Questions:**

1. Could the authors elaborate on the configuration settings for generation, such as temperature and other inference parameters, and how these variances impact performance?
2. Is the benchmark sufficiently challenging given that a 5-shot setup reportedly achieves over 88.68% performance on Qwen?
3. How does GPT-4o perform when incorporating image-based inputs, and could this provide further insights?
4. Since the DavarOCR model is used to extract content information, would employing a more advanced OCR model yield improved results, particularly in light of current performance?

---

### Official Review · Reviewer_Ny59 · 2024-11-07

**Soundness:** 3
**Presentation:** 3
**Contribution:** 3
**Rating:** 6
**Confidence:** 3

**Summary:**

The paper proposes a Chinese document VQA dataset of various domains and question types. The authors conduct a comprehensive analysis of both traditional and LVLM models. The authors also introduce a baseline model with the in-context learning and RAG technique. The method achieves competitive performance in the proposed benchmark.

**Strengths:**

+ The proposed benchmark focuses on the Chinese DocVQA scenarios, filling the blank of multiple domains Chinese VQA benchmark.
+ The benchmark is of a relatively large scale, which encompasses 34k questions. It requires numerous efforts to collect and verify this amount of data.

**Weaknesses:**

+ The proposed baseline method is only competitive for MDCD-VQA. Since the method needs to retrieve TopK similar samples from the training set, if the question image does not fall into the categories in MDCD-VQA, the retrieved samples will be less helpful.
+ Human performance of the proposed benchmark should be included for better reference.

**Questions:**

+ Is the automated question-generation process able to process questions requiring aggregating information from multiple regions in the image?
+ The Equation 1 is a little confusing. The feature of images is not in the same space as the feature of questions. Is there any theory illustration to validate that simply adding these two kinds of features together could represent the example?

---

### Note · Authors · 2024-11-15

I have read and agree with the venue's withdrawal policy on behalf of myself and my co-authors.